# The Processing Differences between Chinese Proper Nouns and Common Nouns in the Left and Right Hemispheres of the Brain

**DOI:** 10.3390/brainsci13030424

**Published:** 2023-02-28

**Authors:** Zijia Lu, Xuejun Bai

**Affiliations:** 1Key Research Base of Humanities and Social Sciences of the Ministry of Education, Academy of Psychology and Behavior, Tianjin Normal University, Tianjin 300074, China; 2Faculty of Psychology, Tianjin Normal University, No.393, Binshui West Street, Tianjin 300387, China

**Keywords:** proper nouns, common nouns, visual half-field technique, lateralization of the brain

## Abstract

In this study, we investigated whether there were differences between the processing of Chinese proper nouns and common nouns in the left and that in the right hemispheres of the brain by using a visual half-field technique. The experimental materials included four types of proper nouns (people’s names, landmark names, country names, and brand names), four types of common nouns (animals, fruits and vegetables, tools, and abstract nouns), and pseudowords. Participants were asked to judge whether target words that had been quickly presented in their left or right visual field were meaningful words. The results showed that there was a distinction between the processing of the two types of words in the left and right hemispheres. There was no significant difference in the processing of the two types of nouns in the right hemisphere, but the left hemisphere processed common nouns more effectively than proper nouns. Furthermore, the processing difference of proper nouns between the two hemispheres was less than that of common nouns, suggesting that proper nouns have a smaller lateralization effect than common nouns.

## 1. Introduction

Nouns can be divided into proper nouns and common nouns according to their contents. The semantic status of the two sorts of words primarily illustrates how they differ from one another. ‘Common noun’ is the name of a class, which means that the semantic meaning of all of its items is the same. ‘Proper noun’, in contrast, is the name of a specific person or thing within a certain class that refers directly to a unique entity. Proper nouns can relate to individuals, animals, stars, geographical features, or distinctive man-made creations such as structures and vehicles [1].

According to evidence regarding patients with aphasia and brain damage, the neural mechanisms involved in processing proper and common nouns overlap and diverge. It is well-known that the left hemisphere (LH) processes words that are presented visually more effectively than the right hemisphere (RH) [2,3]. The left temporal polar (LTP) region plays an important role in the naming of proper and common nouns, according to a variety of studies [4,5,6]. This is where the natural mechanisms overlap. However, it has been discovered that people with severe aphasia and extensive LH impairment still retain the ability to recognize proper nouns. This raises questions about the special role of the RH in processing proper nouns, which is one of the two types of word processing. For instance, it has been shown that people with global aphasia can still recognize the names of famous people when they are written or spoken, despite having difficulty in understanding common nouns [7,8]. In addition, additional investigations have discovered that people with left hemispheric brain damage were nevertheless able to recognize the names of people or places, both verbally and in writing [9,10].

Numerous experiments involving healthy participants have been carried out by researchers to further explore the notion that proper nouns are successfully processed by an intact RH. These studies used the visual half-field technique to examine how the left and right hemispheres of the brain process proper nouns. The human visual nerve conduction pathway’s semicrossover feature is utilized by this technique to investigate the lateralization of stimulus processing between the left and right brain [11]. A stimulus (with a 100–150 ms presentation duration) is quickly delivered to one side of the subject’s visual field while the subject is fixated in the middle of the visual field. Any input can immediately enter the subject’s contralateral cerebral hemisphere from one side of the visual field. In the studies by [12,13], the participants were required to determine whether the target word was a proper noun or a common noun. The findings indicated that while there was no significant difference between the two hemispheres in terms of the accuracy for identifying proper nouns, that for common nouns showed a greater response accuracy in the left hemisphere. The authors claimed that unlike common nouns, which are dominated by the left hemisphere, proper nouns exhibit no such hemispheric dominance, and that the right hemisphere might process proper nouns in a way that is distinct from that of common nouns.

Although the participants could still make classification decisions using other strategies even when they were unfamiliar with the name or word, this prevented the classification task from demonstrating the hemispheric difference that exists between name recognition and word recognition. Schweinberger et al. [14,15] improved on this by directly comparing the hemispheric asymmetry between name recognition and common noun recognition, to determine if the extent of this lateralization varies between the recognition of names and that of common nouns. In the two experiments, participants were asked to determine whether common nouns were real words, and whether proper nouns were well-known. The results contradicted the idea that the right hemisphere has a special role in the processing of proper nouns since they revealed that the accurate rate and reaction time of the two types of words were the same, and the effect magnitude was comparable. 

Although applying the method of Schweinberger et al. [14,15] may be more reasonable, it has one shortcoming. It does not examine how the hemispheric differences between the processing of proper and common nouns interact within an experiment, but rather compares the lexical judgment performances of common and proper nouns between the two experiments. This is beacuse proper nouns must adhere to orthographic norms when they are presented in an alphabetic language writing system (the first letter of proper nouns needs to be capitalized). Some studies have found that the orthographic representation of words contains the case information of the first letter. Words are processed more efficiently in common forms (all lowercase for common nouns, and of the first letter capitalized for proper nouns) than when presented in uncommon forms [16,17,18]. As a result, in the study of alphabetic language, researchers cannot avoid the influence of orthographic rules on lexical processing when they present proper nouns and common nouns in all caps. Furthermore, presenting the two types of words in their common forms adds additional cues that confuse the processing performances of the words themselves. Therefore, the researchers simply compared the processing patterns of the two groups after examining the performance of proper nouns vs. common nouns in the two groups. By this way, they were unable to identify any clear differences in the way that the two types of words are processed in the visual field. 

Chinese is a favorable language with which to the processing traits of proper nouns because it is a nonalphabetic writing script that contains no spelling distinction between proper nouns and common nouns. The present study explores this issue in Chinese using the paradigm proposed by Schweinberger et al. [14,15]. It improves on this paradigm, however, in a number of areas, including: (1) avoiding the impact of orthography on lexical processing andfocusing the processing difference between the two types of words on differences in semantic status. (2) In previous studies, people’s names are often used as proper nouns. In order to draw a broad conclusion, this study broadens its focus to include four types of names: people’s names, landmark names, country names, and brand names. (3) In this study, a linear mixed model is employed to control subjects and items as random effects. Moreover, the familiarity and concreteness of the words were utilized as covariates, which makes the results more trustworthy. 

We hypothesize that there may be a distinction between the processing of the two types of words in the left and right hemispheres if there is considerable interaction between proper nouns and common nouns in the two hemispheres’ processing of the words. Otherwise, the processing mode is comparable.

## 2. Method

### 2.1. Participants

Thirty-two college students (20 women) aged 18 to 22 (*M* = 19.8, *SD* = 1.39) from Tianjin Normal University participated in the experiment. All were right-handed and native Chinese speakers. They reported normal or corrected-to-normal visual acuity, and they had no obvious dyslexia. They were paid after completing the experiment.

### 2.2. Materials

The experimental materials used in research on alphabetic languages typically involve words that range from 4 to 7 letters, and the angle of view is similar to that of two-character and three-character Chinese words. Additionally, according to the general table of modern Chinese [19], the percentage of single-character words is 3%, the percentage of two-character words is 64%, the percentage of three-character words is 18%, and the remainder represents four-character words. The majority of words are two-character and three-character words. Therefore, we used two-character and three-character Chinese words as experimental materials.

#### 2.2.1. Common Nouns

A total of 172 two-character common nouns and 172 three-character common nouns were selected from the Modern Chinese Dictionary [20], including representatives from the four categories of animals, fruits and vegetables, tools, and abstract nouns. We matched the target words’ stroke number and familiarity, selecting words with a stroke number to within ±3 standard deviations and a familiarity to within ±1 standard deviation. The properties of the common words selected were as follows: the stroke numbers of two-character words ranged from 13 to 21, and the stroke numbers of three-character words ranged from 17 to 33, while the familiarity of two-character words ranged from 3.75 to 6.4, and the familiarity of three-character words ranged from 4.5 to 6.2. The familiarity score was derived from the scores of 20 college students. According to their life experience, the familiarity degree of each word was evaluated on a seven-point scale, with 1 indicating “very unfamiliar” and 7 indicating “very familiar”. Furthermore, another 20 college students rated the concreteness of the two types of words. The concreteness degree of each word was evaluated on a seven-point scale, with 1 indicating “high abstractness“ and 7 indicating “high concreteness “. Note, participants who took part in the material evaluation did not participate in the subsequent formal experiment. Finally, 80 common nouns, among two-character and three-character words were selected.

#### 2.2.2. Proper Nouns

A total of 157 two-character proper nouns and 168 three-character proper nouns, including famous people’s names, landmark names, country names, and brand names, were generated through free association by the researchers and five college students. We also controlled for the number of strokes and the familiarity of proper nouns in a similar way as we did for common nouns. The properties of selected proper nouns were as follows: the stroke numbers of two-character words ranged from 13 to 22, and the stroke numbers of three-character words ranged from 16 to 30, while the familiarity of two-word words ranged from 3.75 to 6.25, and the familiarity of three-word words ranged from 4.25 to 6.15. Finally, 80 proper nouns from two-character and three-character words were selected.

#### 2.2.3. Pseudowords

The pseudowords were 80 two-character strings for two-character nouns, and 80 three-character strings for three-character nouns. Each character in strings was a real single-character Chinese word, but when they were joined together, they did not form a meaningful concept. The characters of pseudowords for common nouns and proper nouns were derived from other common nouns and proper nouns, that were not used as target words in this experiment. The number of strokes for the pseudowords used in this experiment matched that for common nouns and proper nouns.

#### 2.2.4. Final Material

There were 160 proper nouns (80 two-character words and 80 three-character words), 160 common nouns (80 two-character words and 80 three-character words), and 160 pseudowords (80 two-character words and 80 three-character words). There was no significant difference among proper nouns, common nouns, and pseudowords in terms of stroke numbers (*F* (2, 479) = 0.26, *p* = 0.774). As for familiarity, there was a significant difference among these three types of nouns (*F* (2, 479) = 4583.64, *p* < 0.001), and the familiarity of common nouns was higher than that of proper nouns (*t* (159) = −9.89, *p* < 0.001). The mean and standard deviation of stroke numbers and the familiarity of target words are shown in Table 1. We opted to control for nonsignificant differences in the number of strokes, and then controlled for nonsignificant differences in familiarity as much as possible. However, there was a significant difference in familiarity between common nouns and proper nouns. Therefore, familiarity was controlled for as a covariate in the results.

### 2.3. Apparatus

The experimental materials were presented using E-Prime version 2.0, and the materials were presented on the laptop screen with a screen resolution of 1366 × 768 and a screen refresh rate of 60 Hz. The subjects’ eyes were centered 70 cm from the computer screen, facing squarely toward the center of the display. The target word was set to Song font at font size 32 as white words on a black background, corresponding to a character height of 1.1 cm and a visual angle of 0.9°. The length of the two-character word was 2.2 cm, corresponding to a visual angle of between 2.1° and 3.9°. The length of the three-character word was 3.2 cm, corresponding to a visual angle of between 1.7° and 4.3°. The distance between the middle position of the word and the center position of the fixation point “+” was 3.7 cm, corresponding to a visual angle of 3°. The height of the chin rest was adjusted according to the height of the subjects, so that the subjects could complete the experiment in a comfortable posture. The chin rest was used to prevent subject head movements during the experiment, ensuring that the target word appears in either the left or right visual field.

### 2.4. Procedure

The experiment was carried out in a relatively quiet laboratory environment. Each participant was tested separately under the guidance of the experimenter. The participants first completed the informed consent form and then placed their chin on a chin rest. The experimenter displayed the instructions to participants and helped explain the process of the experiment as follows: “You need to carry out the following two tasks for this experiment: Pay close attention to the ‘+’ when it appears in the middle of the screen, and then decide whether the quickly presented string to the left and right of the ‘+’ is a meaningful word. Both speed and accuracy are emphasized.”

In each trial, the “+” was first presented in the centre of the screen for 500 ms, and then a target word or pseudoword was randomly presented to the left (LVF) or right (RVF) of the “+” for 150 ms. A string of # with an average length that matched the average length of the target stimuli was presented in the region that was contralateral to the target stimulus. This was performed to maximize the possibility that the centre of attention would be the focus [21]. Then, a blank screen with the longest duration of 2000 ms was displayed. To determine whether the word on the previous screen was a meaningful word, the participants could press buttons on this screen. The screen ended after pressing the key or after reaching 2000 ms. The 1500 ms feedback was then presented in the centre of the screen with a green “Correct” for a correct response and a red “Error” for an incorrect response. Before the experiment, 16 practice trials were run. To participate in the formal experiment, the accurate rate duringpractice had to approach 80%. Otherwise, the practice was repeated. The experimental flow chart is shown in Figure 1.

The target words in this experiment were split into two sections, “A” and “B,” which each comprised 240 words (80 proper nouns, 80 common nouns, and 80 pseudowords), which were displayed in the left visual field and the right visual field, respectively. In the two blocks, either “A” was shown on the right and “B” was presented on the left, or vice versa. In addition, half of the participants pressed “F” if they thought it was a meaningful word (“YES”) and pressed “J” if they thought it was a pseudoword (“NO”). In contrast, the other half of the participants pressed “J” for “NO” and “F” for “YES”. Therefore, there were four blocks of eight valid participants. The duration of this experiment was a 40 min.

## 3. Results

An accuracy rate of more than 75% for 32 participants showed that they were well-trained in the experimental task and careful in their execution. Trials with a reaction time of less than 300 ms were eliminated due to the potential for systematic errors (machine errors) or participant carelessness. Based on this threshold, 480 of the trials for two-character terms and 367 of the trials for three-character terms were disregarded. Trials were removed if the reaction times were above or below 3 SDs from each participant’s mean. Based on this threshold, 80 trials for two-character terms and 72 trials for three-character terms were disregarded. We deleted 999 trials in total, accounting for 5% of the total data volume.

We analyzed the accuracy for all responses and response times for correct responses. For data analysis, we executed a linear mixed-effects model (LMM) for response times, and a generalized linear mixed-effects model (GLME) for response accuracy in the R environment [22] (Version 4.1.2) using the lme4 package [23]. The model took fixed effects into account, including visual field, the classification of nouns, and their interactions. Additionally, participants and items were entered as crossed random effects. The random effects structure of the model was determined by starting with the maximal random effects structure [24]; however, if the maximum random model did not converge, it was further trimmed down. The response times were log-transformed to normalize the distribution before analysis. In addition, the familiarity of the target word was analyzed as covariates.

Table 2 shows the mean and standard deviations for response accuracy and response times, and Table 3 shows the corresponding fixed effect estimations. The comparisons of the average difference of response accuracy are shown in Figure 2.

### 3.1. The Main Effect of the Visual Field

The participants were faster (*b* = −0.03, *SE* = 0.01, *t* = −2.27, *p* = 0.027, 95%CI = [−0.06, 0]) and responded more accurately (*b* = 0.47, *SE* = 0.13, *z* = 3.49, *p* < 0.001, 95%CI = [0.21, 0.73]) at judging whether the target word that appeared in the RVF was a meaningful word than they did when appeared in the LVF. This indicated that the left hemisphere (RVF) processed words better than the right hemisphere (LVF).

### 3.2. The Main Effect of the Classification of Nouns

The response accuracy of common nouns was significantly higher than that of proper nouns (*b* = −0.34, *SE* = 0.13, *z* = −2.62, *p* = 0.009, 95%CI = [−0.6, −0.09]), and the response times of common nouns were significantly shorter than that of proper nouns (*b* = 0.04, *SE* = 0.01, *t* = 4.03, *p* < 0.001, 95%CI = [0.02, 0.06]), suggesting that proper nouns were more difficult to process than common nouns.

### 3.3. Interaction between Visual Field and the Classification of Nouns

There was a significant interaction between the visual field and the classification of nouns in response accuracy (*b* = −0.34, *SE* = 0.15, *z* = −2.23, *p* = 0.026, 95%CI = [−0.63, −0.04]). Follow-up GLMMs revealed that the response accuracy to proper nouns was equal to that of common nouns in the LVF (*b* = 0.13, *SE* = 0.13, *z* = 1.02, *p* = 0.306, 95%CI = [−0.12, 0.39]). However, when the target words were presented in the RVF, participants responded more accurately to common nouns than to proper nouns (*b* = 0.44, *SE* = 0.13, *z* = 3.37, *p* < 0.001, 95%CI = [0.19, 0.70]). On the other hand, the difference in the processing of proper nouns between the LVF and RVF (*b* = 0.27, *SE* = 0.09, *t* = 2.91, *p* = 0.004, 95%CI = [0.09, 0.45]) was numerically smaller than the difference in processing of common nouns between the LVF and RVF (*b* = 0.65, *SE* = 0.10, *t* = 6.40, *p* < 0.001, 95%CI = [0.45, 0.85]). In terms of response times, there was no significant interaction between visual field and the classification of nouns (*b* = 0.00, *SE* = 0.01, *t* = 0.48, *p* = 0.631, 95%CI = [−0.01, 0.02]).

### 3.4. Further Analysis

After controlling the familiarity as a covariate, the patterns did not change much regarding the main effect of the visual field (response accuracy: *b* = 0.48, *SE* = 0.08, *z* = 6.79, *p* < 0.001, 95%CI = [0.34, 0.61]; response times: *b* = −0.03, *SE* = 0.01, *t* = −2.32, *p* = 0.024, 95%CI = [−0.06, 0]), and the interaction between the visual field and the classification of nouns (response accuracy: *b* = −0.43, *SE* = 0.14, *z* = −3.09, *p* = 0.002, 95%CI = [−0.7, −0.16]; response times: *b* = 0, *SE* = 0.01, *t* = 0.46, *p* = 0.647, 95%CI = [−0.01, 0.02]). However, the difference between common nouns and proper nouns became nonsignificant for response accuracy (*b* = 0.01, *SE* = 0.14, *z* = 0.07, *p* = 0.94, 95%CI = [−0.26, 0.28]) and response times (*b* = 0, *SE* = 0.01, *z* = 0.28, *p* = 0.78, 95%CI = [−0.02, 0.02]) after controlling familiarity as a covariate. This indicated that the processing difference between the two types of words may be mainly caused by the familiarity. Fixed effect estimations for the response accuracy and response times with familiarity as a covariate are shown in Table 4.

In addition, we also analyzed the concreteness as a covariable. After controlling the concreteness as a covariate, the patterns had not changed much on the main effect of the visual field (response accuracy: *b* = 0.48, *SE* = 0.07, *z* = 6.82, *p* < 0.001, 95%CI = [0.34, 0.61]; response times: *b* = −0.03, *SE* = 0.01, *t* = −2.27, *p* = 0.027, 95%CI = [−0.06, 0]), the classification of nouns (response accuracy: *b* = −0.53, *SE* = 0.14, *z* = −3.83, *p* < 0.001, 95%CI = [−0.80, −0.26]; response times: *b* = 0.04, *SE* = 0.01, *t* = 4.10, *p* < 0.001, 95%CI = [0.02, 0.06]), and interaction between the visual field and the classification of nouns (response accuracy: *b* = −0.43, *SE* = 0.14, *z* = −3.11, *p* = 0.002, 95%CI = [−0.71, −0.16]; response times: *b* = 0, *SE* = 0.01, *t* = 0.49, *p* = 0.628, 95%CI = [−0.01, 0.02]). This indicated that the concreteness of the two types of words had no effect on processing. Fixed effect estimations for the response accuracy and response times with concreteness as a covariate are shown in Table 5.

## 4. Discussion

This study explored the differences between Chinese proper nouns and common nouns in their processing by the left and right hemispheres of the brain by using the visual half-field technique. The results showed that word processing benefited from the left hemisphere of the brain. Additionally, we discovered that the processing of proper nouns and common nouns did not differ significantly. Furthermore, the processing patterns of the two types of words in the left and right hemispheres were not consistent in terms of response accuracy. On the one hand, there was no significant difference in the processing of the two types of nouns in the right hemisphere, but the left hemisphere processed common nouns more effectively than proper nouns. On the other hand, the processing difference of the proper nouns between the two hemispheres was less than that of the common nouns.

The results of this experiment differ from the related research findings that were discussed in the introduction. There are two possible reasons for this difference. Initially, there are variations in the task requirements and experimental designs across the various studies. In Ohnesorge and Van Lancker’s [12] experiment, the participants were asked to classify common and proper nouns rather than perform a lexical judgement task. That is, the participants were instructed to “judge whether the target word is a proper noun or a common noun,” “judge whether the proper noun is female or male and whether the common noun is inanimate or animated,”, etc. Instead of completing a classification task, the participants in this study completed a judgement task. They were asked to judge whether the presented string was a meaningful word. The task was the same as in the study of Schweinberger et al. [14,15], but the experimental design was not. Schweinberger et al. [14,15] separately compared familiar names with unfamiliar names and common nouns with false words, and then compared whether there was any difference between the two processing modes and the effect size. The estimated value of the interaction may be directly obtained in this experiment. The current study’s findings confirm Ohnesorge and Van Lancker’s [12] view that proper nouns are processed similarly in both hemispheres, but they do not confirm the theory that proper nouns are more effective than common nouns. It is impossible to directly compare the two types of task requirements, since the mechanisms underlying them are unknown. Further studies are required to investigate the processing variations between categorical and judgmental tasks. The second reason is the development of analysis techniques and the management of unimportant variables. Previous studies have had heated discussions about whether familiarity was sufficiently controlled in earlier investigations. In this study, familiarity was first controlled by evaluation and selection within a comparable interval, and then, secondary management was handled using the statistical approach as a covariable. It is clear that familiarity had a substantial impact on the results. In addition to the control of familiarity, this study adopts a linear mixed model to control the subjects and material variables as random variables, and to control the concreteness of target words as a covariable, ensuring that the influence of extraneous variables is minimized. The experimental findings are trustworthy.

We mostly discovered the following three significant findings in this investigation.

First, since the middle of the 19th century, it has been well acknowledged that the left hemisphere predominantly governs the functions associated with language. Studies on brain-damaged patients have revealed that Broca’s area, which performs language production, and Wernicke’s area, which performs language comprehension, are both controlled by the left hemisphere of the brain. This is supported by the fact that words are processed in the left hemisphere in the current experiment.

Second, the processing difference between proper nouns and common nouns becomes nonsignificant after controlling the familiarity of proper nouns and common nouns as covariates, suggesting that familiarity may be the source of the processing difference. Word frequency has been discovered to have a significant impact on how common nouns are processed in studies. Shorter fixation times and better reaction accuracy are both associated with high frequency words [25,26,27,28]. In this study, we were unable to find a reliable text corpus that contained information on the word frequency of both proper and common nouns in Chinese. Proper nouns typically have low word frequencies because they are less common in normal conversation, even if such a corpus does exist. Therefore, we decided to utilize familiarity rather than word frequency to control the difficulty in processing the two different kinds of words, and we chose target words with a similar level of familiarity as our experimental materials. We tried our best to balance two additional factors, namely, stroke numbers and familiarity, while choosing materials, but we were still unable to achieve optimal control. This is beacuse the number of typically common nouns is limited, and the stroke numbers of words with similar levels of familiarity can vary substantially, which could significantly affect the processing load. On the basis of controlling the nonsignificant difference in the number of strokes, we selected the target words within 1 standard deviation of the average of familiarity scores. However, there was still a significant difference in familiarity scores between proper nouns and common nouns. After evaluation by participants, the average familiarity scores of proper nouns and common nouns were 5.3 and 5.92 (seven-point score), respectively. The additional variable of familiarity was controlled as a covariate in the analysis. The results showed that familiarity affected the processing performances of the two types of words. After controlling for familiarity, no significant difference was found between the two types of words. This indicates that for healthy subjects, the two types of word processing have basically the same performances in behaviour indicators.

Third, there was a significant interaction between visual field and word type in response accuracy. The two types of words were processed differently in the left and right hemispheres, according to subsequent analysis.

On the one hand, proper nouns and common nouns are processed similarly in the right hemisphere, whereas common nouns are processed considerably more easily in the left hemisphere than proper nouns. The higher analytical ability of the left hemisphere, which makes it easier to process common nouns, could be a possible reason. Sperry’s research on split-brain patients found that the left hemisphere was mostly responsible for logical and analytical thinking, while the right hemisphere was primarily responsible for intuitive and comprehensive thinking. The processing of common nouns requires the ability to use abstract logical reasoning, while the processing of proper nouns relies on this less. The storage of common nouns in the brain is constructed in the form of networks and nodes. Collins et al. proposed applying the hierarchical network model [29] and the spreading activation model [30]. In the two models, the layers from the top level (abstract) to the bottom level (concrete), and the length of the line representing the similarity of concepts reflect the importance of lexical semantic meaning in the network. For common nouns, an upper concept contains several lower concepts, and one concept can activate many other concepts that are related to its semantic meaning, which can provide certain characteristic information. However, there is no similar semantic network for proper nouns to represent conceptual information. The absence of description for proper nouns corresponds to meaninglessness. In other words, proper nouns do not attribute any characteristics to the referred entity [31,32,33]. The difference in semantic status directly results in the slow retrieval speed of proper nouns, and once the connection of this one-to-one relationship is broken, the word will not be recovered [34,35]. Burke et al. (1991) proposed an interaction activation model that incorporates the idea that proper nouns do not include conceptual information [31]. Compared to the connection of common nouns between propositional nodes (which represent the conceptual information and semantic attributes of the category) and lexical nodes, proper nouns contain an additional intermediary stage in their processing, which is called the “proper noun phrase.” This means that for proper nouns, the stimulus is only delivered to the lexical node through a single connection during the “proper noun phrase,” and additional top-down connections are unavailable to compensate for faulty transmission. Common nouns are made up of hierarchical concepts, whereas proper nouns need to make direct references to specific individuals rather than having to deal with complicated semantic information. Common nouns are more likely than proper nouns to benefit from the superior analytical ability of the left hemisphere while word processing. Therefore, their processing gains significant advantages in the left hemisphere.

On the other hand, the processing difference of proper nouns between the two hemispheres is smaller than that of common nouns, meaning that proper nouns have a smaller lateralization effect than common nouns. In addition to the left hemisphere’s unique promotion of common nouns, another possible reason is that the right hemisphere may also have a unique capability with proper nouns, thanks to the right hemisphere is ability to process familiar stimuli. Familiarity agnosia is known to be associated with RH impairment [36,37,38]. Patients with severe aphasia, whose right hemisphere is intact, perform better on stimulus tasks involving personal familiarity [39]. In a split-brain patient study, patients with RH damage were found to have successfully identified personally relevant stimuli [40]. Studies on behaviour, neurophysiology, and neuroimaging in healthy people have also indicated widespread right hemispheric lateralization in the recognition offamiliar faces. Kloth et al., observed a clear modulation of M170 by familiar faces, which evoked greater amplitudes in the right hemisphere than unfamiliar faces [41]. Sun et al. (2012) found that familiar faces evoked stronger negative waves at between 300 and 500 ms (N400f) in the right parietal and temporal regions than unfamiliar faces [42]. Eger et al., evaluated prime-related repetition effects on fMRI using both well-known and unfamiliar faces [43]. They discovered that the repetition of well-known rather than unfamiliar faces resulted in a larger drop-in activity in the right anterior fusiform gyrus region. In the current experiment, we used proper nouns that the participants were familiar with, which may be regarded as a “meaningless” label in left hemisphere processing, but which may be regarded as an object that evokes cognitive and emotional familiarity in right hemisphere processing, leading to increased right hemisphere involvement.

We can further examine how various forms of proper nouns are processed differently in the future. In this study, proper nouns from four different categories are employed. Our simple analysis found that country names had the highest processing accuracies, and the quickest reaction times. People’s names and landmark names came in second. Brand names were last. This suggests that the processing of various kinds of proper nouns differs in several ways. The current study did not fully explore and examine the problem; thus, future discussions should explore this in more detail.

## 5. Conclusions

In this study, we investigated the processing of Chinese proper nouns and common nouns in the left and right hemispheres using the visual half-field technique. We found the following:

(1) Word processing benefited from the left hemisphere of the brain;

(2) The processing of proper nouns and common nouns did not differ significantly;

(3) There was no significant difference in the processing of the two types of nouns in the right hemisphere, but the left hemisphere processed common nouns more effectively than proper nouns; and

(4) The processing difference of proper nouns between the two hemispheres was less than that of common nouns, suggesting that proper nouns have a smaller lateralization effect than common nouns.

## Figures and Tables

**Figure 1 brainsci-13-00424-f001:**
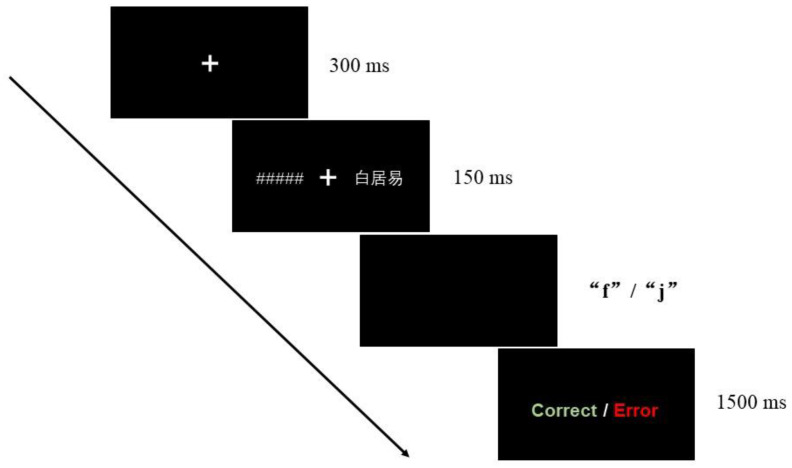
The experimental process. A string of # with an average length that matched the average length of the target stimuli was presented in the region that was contralateral to the target stimulus. “白居易” is the name of a famous Chinese poet in the Tang Dynasty. The feedback was presented in the centre of the screen with a green “Correct” for a correct response and a red “Error” for an incorrect response.

**Figure 2 brainsci-13-00424-f002:**
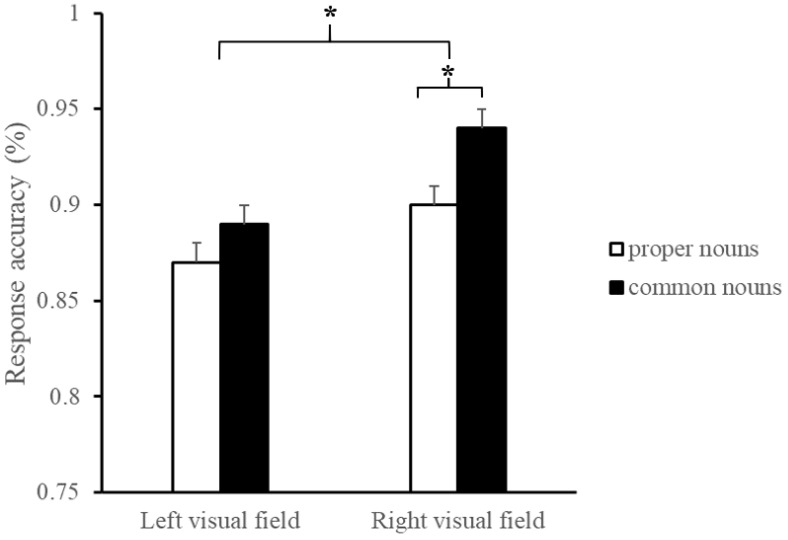
Mean response accuracy for all conditions. *Error bars*: *SE* of the mean. ”*” means the differences reach significant.

**Table 1 brainsci-13-00424-t001:** Properties for target words.

	Number	Stroke Numbers		Familiarity		Concreteness	
		*M*	*SD*	*M*	*SD*	*M*	*SD*
Proper nouns	160	19.93	4.78	5.30	0.54	6.20	0.28
Common nouns	160	20.29	4.72	5.92	0.49	5.01	0.18
Pseudowords	160	20.04	4.61	1.20	0.40		

**Table 2 brainsci-13-00424-t002:** Means and standard deviations (in parentheses) for each condition.

	Response Accuracy		Response Times	
	Left Visual Field (LVF)	Right Visual Field (RVF)	Left Visual Field (LVF)	Right Visual Field (RVF)
Proper nouns	0.87 (0.08)	0.90 (0.07)	517 (86)	503 (83)
Common nouns	0.89 (0.08)	0.94 (0.05)	500 (93)	480 (76)

**Table 3 brainsci-13-00424-t003:** LMM Analyses for Response Accuracies and Response Times.

	Response Accuracy			Response Times		
	*b*	*SE*	*z*	*b*	*SE*	*t*
Intercept	2.47	0.08	**32.52**	6.18	0.02	**346.62**
Visual field (V)	0.47	0.13	**3.49**	−0.03	0.01	**−** **2.27**
Classification of nouns (C)	−0.34	0.13	**−** **2.62**	0.04	0.01	**4.03**
V × C	−0.34	0.15	**−** **2.23**	0.00	0.01	0.48
a	0.13	0.13	1.02			
b	0.44	0.13	**3.37**			
c	0.27	0.09	**2.91**			
d	0.65	0.10	**6.40**			

*Note.* Significant items are presented in bold. *b* = regression coefficient. a = the difference between processing of proper nouns and common nouns in the LVF; b = the difference between processing of proper nouns and common nouns in the RVF; c = the processing difference of proper nouns between the LVF and RVF; d = the processing difference of common nouns between the LVF and RVF. Significant effects are indicated in bold.

**Table 4 brainsci-13-00424-t004:** LMM Analyses for Response Accuracy and Response Times, with Familiarity as a Covariate.

	Response Accuracy			Response Times		
	*b*	*SE*	*z*	*b*	*SE*	*t*
Visual field (V)	0.48	0.08	**6.79**	−0.03	0.01	**−** **2.32**
Classification of nouns (C)	0.01	0.14	0.07	0.00	0.01	0.28
Familiarity	0.57	0.09	**6.32**	−0.05	0.01	**−** **8.71**
V × C	−0.43	0.14	**−** **3.09**	0.00	0.01	0.46

*Note.* Significant items are presented in bold.

**Table 5 brainsci-13-00424-t005:** LMM Analyses for Response Accuracy and Response Times, with Concreteness as a Covariate.

	Response Accuracy			Response Times		
	*b*	*SE*	*z*	*b*	*SE*	*t*
Visual field (V)	0.48	0.07	**6.82**	−0.03	0.01	**−** **2.27**
Classification of nouns (C)	−0.53	0.14	**−3.83**	0.04	0.01	**4.10**
Concreteness	0.14	0.04	**3.26**	0.00	0.00	−0.98
V × C	−0.43	0.14	**−** **3.11**	0.00	0.01	0.49

*Note.* Significant items are presented in bold.

## Data Availability

The data presented of this study are available on request from Corresponding author.

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
