# Peer review of "The Processing Differences between Chinese Proper Nouns and Common Nouns in the Left and Right Hemispheres of the Brain"

_brainsci, 2023, doi:10.3390/brainsci13030424_

Round 1
Reviewer 1 Report
The manuscript entitled "The processing differences between Chinese proper nouns and common nouns in the left and right hemispheres of the brain" described an interesting study to examine the hemispheric lateralization to examine the processing of common and proper nouns. Overall, the manuscript was clearly-written and the study was well-designed. The analytical approach was proper and the conclusion was based on the results. I have the following comments and suggestions for the authors to improve the manuscript.
My major concern is whether the interaction between noun categories and hemispheres is driven by imageability. With mounting evidence that the LH is more for verbal and RH is more for nonverbal and pictorial processing (e.g., studies on semantic dementia with major LH or RH damages), the interaction is possible to be driven by the fact that the proper nouns are highly imageable since they refer to a very specific person, location, etc. The authors could provide additional data or analysis showing whether the results are affected by this factor. Regardless, authors should consider this theoretical account for explaining the results.
The method section should be more clear. The following are the concerns about the methods:
1. Are any of the participants who rated the familiarity of words part of the main cohort for the experiment? If not, please specify.
2. Were any of the response time data of incorrect trials excluded for analysis?
3. Can the authors explain why LMM and GLMM were applied to accuracy and RT analyses separately? What was the rationale? Did the final models include the participants and items as random effects for intercept or slope or both?
Minor concerns:
1. The 2nd paragraph of the intro seems to have contradictory statements, People with LTP damage had trouble in both common and proper nouns, vs. people with extensive LH impairment still retain the ability to recognize the proper nouns.
2. The instructions part for the method and results were included in the main text.
3. Careful language checking is needed.
Reviewer 2 Report
It is unclear what the theoretical basis of this study is, as there is no separate section devoted to it. In the discussion section, it is better to list all the similarities and differences found between this and other studies, based on the theoretical section inserted during the revision.
Author Response
Please see the attachment.
We have sought the help of AJE before submitting the manuscript. If you think the language fluency still needs improvement, we can continue to improve it.

Round 2
Reviewer 2 Report
The changes made are sufficient and well-reasoned in the authors' response. Thank you for your work!